# Heterogeneous Graph Guided Contrastive Learning for Spatially Resolved Transcriptomics Data

## ABSTRACT

Spatial transcriptomics provides revolutionary insights into cellular interactions and disease development mechanisms by combining high-throughput gene sequencing and spatially resolved imaging technologies to analyze genes naturally associated with spatially variable tissue genes. However, existing methods typically map aggregated multi-view features into a unified representation, ignoring the heterogeneity and view independence of genes and spatial information. To this end, we construct a heterogeneous **G**raph guided **C**ontrastive **L**earning (stGCL) for aggregating spatial transcriptomics data. The method is guided by the inherent heterogeneity of cellular molecules by dynamically coordinating triple-level node attributes through comparative learning loss distributed across view domains, thus maintaining view independence during the aggregation process. In addition, we introduce a cross-view hierarchical feature alignment module employing a parallel approach to decouple spatial and genetic views on molecular structures while aggregating multi-view features according to information theory, thereby enhancing the integrity of inter- and intra-views. Rigorous experiments demonstrate that stGCL outperforms existing methods in various tasks and related downstream applications.

## CCS CONCEPTS

• **Computing methodologies** → **Knowledge representation and reasoning**; **Discrete space search**; • **Mathematics of computing** → **Information theory**.

## KEYWORDS

multi-view learning, cluster, cross-modality

## 1 INTRODUCTION

Spatial transcriptomics is widely employed to explore the structural organization and functional roles of cells within tissues. This approach seeks to elucidate the spatial patterns of gene expression critical for applications, e.g., precision medicine, resolution of disease mechanisms, and discovery of disease biomarkers [3]. Unlike single-cell multi-omics, spatial transcriptomics offers insights into the spatial arrangements of cells, facilitating new perspectives on cell-cell interactions in complex biological processes [2, 31].

Top performers have merged spatial transcriptomics with computational vision, mathematical models, and graph convolution to

*MM '24, Melbourne, Australia.,*
© 2024 Association for Computing Machinery.
ACM ISBN 979-8-4007-0108-5/23/101...$15.00

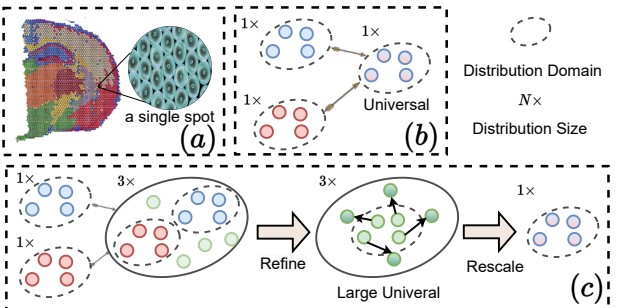

**Figure 1: (a) A spot in spatial transcriptomics indicates multiple cellular genes. (b) Typical methods aggregate multi-modal data into a compact universal distribution. (c) Our proposed mapping-to-expansion paradigm maximizes the preservation of the original modal molecular information.**

identify spatially variable genes, intending to leverage morphological priors of spatial features for cellular discrimination [6, 35]. However, organisms are defined by the relationships between cellular genes and cells with a regular spatial distribution in morphology [19]. It is quite a challenging task to combine the best of both perspectives, i.e., to integrate spatial and genetic information based on a priori relationships [21].

Recent proposals for graph neural networks (GNNs) to integrate spatial and genetic information, thereby processing multi-modal data from various sensors and learning to discern associations between modalities and adaptively learn associations between modalities [23]. SpaGCN [9] aggregates gene expression around each spot through spatial graph convolution. DeepST [29] integrates node features and positional information using denoising autoencoders and GNNs. Li et al. [11] introduces an unsupervised cell clustering method based on graph convolutional networks. GraphST [14] further integrates multiple tissue sections vertically and horizontally through spatial graph self-supervised contrast. MuSe-GNN [13] combines weighted similarity and contrastive learning for regularization, aiming to learn gene-gene relationships across datasets. Furthermore, due to sensor limitations, genetic data for one locus typically contains an average of more than a dozen cellular expressions [4]. Thus, the independence of gene and spatial views of cells is particularly critical for unbiasedly determining cell population boundaries for categorization, i.e., returning to the basics.

As shown in Fig. 1, a cellular organization typically exhibits spatial regularity resulting from cell expansion processes. However, the aforementioned method ignores the potential heterogeneity of spatial and gene associations for the expression of cellular categories, resulting in the absence of inter-cellular associations during the fusion process. There are two points: **(1)** Although spatial features

at the boundaries of cell populations are highly correlated, cell categorization remains inconsistent and primarily dependent on gene markers. This inconsistency can cause spatial features to overly smooth representations at tissue boundaries, leading to confusion in cell categorization. **(2)** From a multi-view alignment perspective, previous studies have shown that fusion map distributions were equivalent in size to the distributions of individual modalities, obscuring the structural organization of spatial and genetic features and contributing to modality collapse during the fusion process (as depicted in Fig. 1). This is like shooting arrows; the larger the target (i.e., for the projected feature distribution), the better the arrow shot from each modality hits the bull's eye.

To address the above issues, we propose an end-to-end framework of heterogeneous graph Guided Contrastive Learning (stGCL) for spatially resolved transcriptomics data. This framework constructs gene and spatial graphs utilizing features derived from independent parallel graph encoders, which are subsequently represented within latent distributions. To identify confusing organizational boundaries, we construct three levels based on spatial and genetic heterogeneity relationships, i.e., the joint, gene, and tissue levels, enabling the unbiased refinement of node attributes through heterogeneous contrastive learning loss, leveraging the independent a priori correlations among the three views.

On the other hand, to alleviate the modality collapse problem, we design a cross-view hierarchical feature alignment (CHA) aggregator designed to integrate cross-view features at the molecular structure level while simultaneously tracing back to identify view-specific structures. Inspired by information theory, the aggregated features are refined to encompass heterogeneous and homogeneous features, resulting in a comprehensive collection of coherent universal features with a consistent distribution size (as depicted in Fig. 1c). stGCL incorporates a priori modeling for each view and implements discriminative clustering techniques ranging from coarse to fine, from tissue to molecular formulae. This integration ensures a genetic representation of the functional similarity of morphology across the views encompassed within the joint space.

**Contributions:** The main contributions of this paper are:

- We propose a stGCL for spatial transcriptomics data that end-to-end combines the heterogeneity of genetic and spatial a priori distributions to learn the intrinsic local organization of cells, providing a novel perspective on the mechanism of cellular interactions to address the coordination between different views in tissue.
- A CHA module is designed to aggregate cross-view features at the structure level, maintaining the independent structural integrity of the spatial and genetic views according to information theory with a mapping-rescale paradigm.
- We introduce Contrastive Heterogeneous Molecular Learning to identify node attributes based on a priori correlations of domain distributions at different levels to recognize molecular latent spaces in an unbiased manner.
- We conduct comprehensive experiments on various three datasets, demonstrating the quantitative and qualitative superiority of our method. Its effectiveness was also confirmed in downstream ST tasks.

## 2 RELATED WORK

### 2.1 Spatial transcriptomics

Spatial transcriptomics (ST) techniques are widely used in various fields, including oncology, neuroscience, and developmental biology [5], providing new insights not available from traditional transcriptomics techniques that lack spatial resolution [4, 27]. Deciphering the spatial organization of gene expression can facilitate the discovery of new cell types [4], delineation of molecular pathways [22], and the identification of targets for therapeutic interventions [33], among other applications [16]. ST has unique spatial organization information relative to single-cell multi-omics, so some excellent work has been done to construct representation learning networks based on spatial graphs to identify spatially variable genes from a spatial perspective and to identify cells using the morphological prior distribution of the spatial features [16]. Giotto employs a Hidden Markov Random Field to model gene expression at nodes, inspired by super-resolution in computer vision. SpaGCN [9] uses GCN to identify spatial domains based on gene expression, histology, and spatial location aggregation, identifying spatially variable genes within each domain. SEDR [30] trains both a depth self-encoder and a self-encoder to learn low-dimensional spatial embedding of ST. Graph self-encoder to learn the low-dimensional spatial embedding of ST. Squidpy [17] integrates omics and image analysis tools to facilitate an expandable description of ST. Inspired by the super-resolution technique in computer vision, BayesSpace [34] has developed a comprehensive Bayesian statistical model incorporating a Markov random field.

### 2.2 Multi-view Graph Learning

Many studies [24] have explored the application of graph neural networks (GNNs) in partitioning and identifying spatial domains. These studies treat spatial and genetic features as distinct sources of multi-view information, aiming to facilitate adaptive learning of inter-view associations [7, 8]. For instance, stLearn [18] employs neighbor-based smoothing and morphological adjustment alongside a graph-based clustering approach to identify spatial domains by normalizing ST data. SpaceFlow [20] introduces spatially regularized deep graph networks to produce spatially coherent low-dimensional embeddings. ThItoGene [10] utilizes dynamic convolutional and capsule networks to detect potential molecular signals in histological images adaptively. Spatial-MGCN [26] employs graph convolution to adaptively learn intricate relationships between gene expression and spatial information. MuSe-GNN [13] integrates weighted similarity learning and contrast learning for regularization, enabling the exploration of gene-gene relationships across datasets. In this study, we propose a multi-view contrast learning framework based on spatial and gene views complemented by a multi-level heterogeneous guided fusion strategy. This framework aims to collaboratively mitigate the mismatch in cross-view feature expression during fusion.

## 3 METHODOLOGY

### 3.1 Overview

Given spatial and gene features, a spatial graph $\psi_{spa} = (\psi_s, X)$ is constructed based on neighbor associations, where $\psi_s \in \mathbb{R}^{N \times N}$

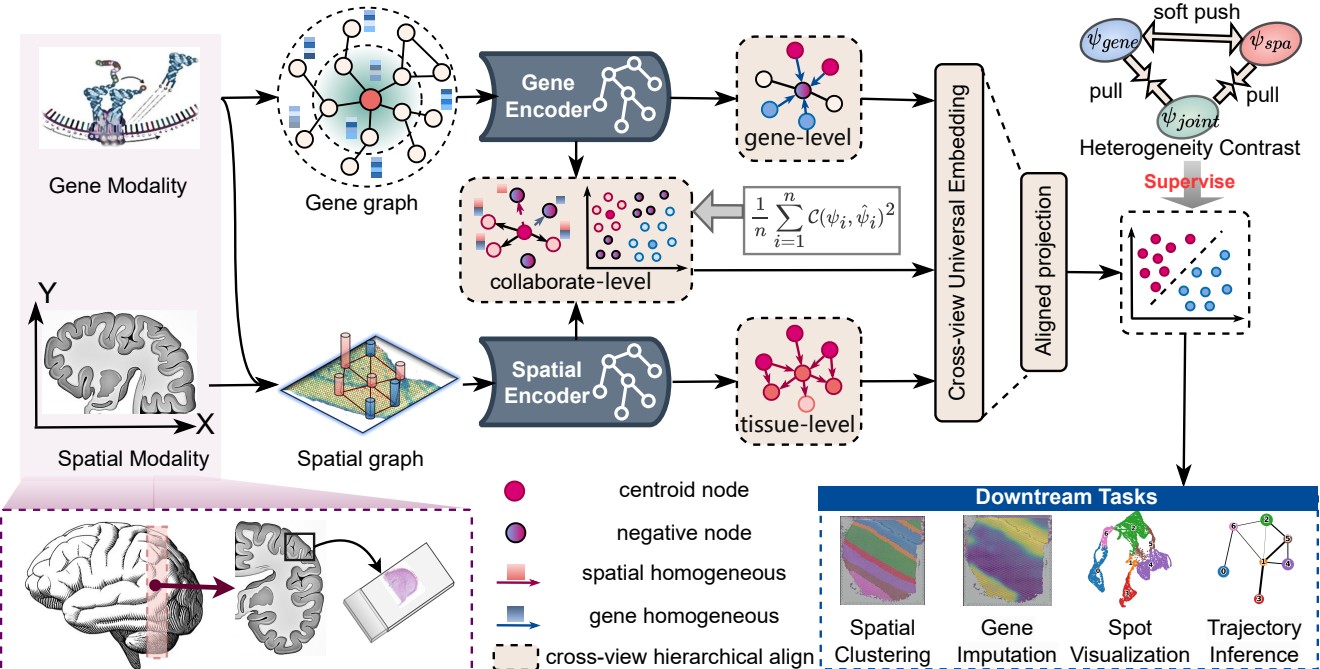

**Figure 2: Illustration of the proposed spatial transcriptomics method stGCL. Cross-view aggregation for global structure awareness of the graph (i.e., a low-pass filter) can filter redundant information from highly sparse biological data. According to information theory, the cross-view generic embedding and alignment projections included in cross-view hierarchical feature alignment (CHA) are designed to maximize the representation of spatially and genetically critical information in the final features for application in various downstream tasks.**

is the spatial adjacency matrix for $N$ points, and $X \in \mathbb{R}^{N \times M}$ represents the normalized gene expression matrix, with $M$ being the number of filtered genes. We set $\psi_s^{ij} = \psi_s^{ji} = 1$ if the Euclidean distance $S_{ij}$ between points $i$ and $j$ is less than the predefined radius $r$, otherwise, it is set to 0, which can be formalized as:

$$\psi_s^{ij} = \begin{cases} 1, & \text{if } S_{ij} \leq r \\ 0, & \text{otherwise.} \end{cases} \quad (1)$$

On the other hand, the gene expression graph is formed by measuring the similarity of gene expressions using cosine distance. Specifically, we construct the k-nearest neighbor graph $\psi_{gene} = (\psi_f, X)$ of the gene expression matrix $X$, where $\psi_f \in \mathbb{R}^{N \times N}$ is the feature adjacency matrix. The adjacency matrix $A_f$ for this graph is defined using a binary classification system, where:

$$A_{ij}^f = \begin{cases} 1 & \text{if } j \text{ is a neighbor of } i, \\ 0 & \text{otherwise.} \end{cases} \quad (2)$$

ensure that only the first $k$ neighbors of each point based on the gene cosine similarity are considered in the graph, and calculated using the gene expression vectors $\mathbf{x}_i$ and $\mathbf{x}_j$ from $X$ to compute the cosine similarity between the two points $i$ and $j$, with the formula: $\text{sim}(\mathbf{x}_i, \mathbf{x}_j) = \frac{\mathbf{x}_i \cdot \mathbf{x}_j}{\|\mathbf{x}_i\| \|\mathbf{x}_j\|}$.

The spatial and gene-view graphs are then processed separately through graph convolutional encoders to maximize their potential

expression, formulated as follows:

$$H^{(l+1)} = \sigma(\tilde{D}^{-\frac{1}{2}} \tilde{\psi} \tilde{D}^{-\frac{1}{2}} H^{(l)} W^{(l)}), \quad (3)$$

where $H^{(l+1)}$ denotes the node embeddings at the $(l + 1)$-th layer, $H^{(l)}$ represents the node features at the $l$-th layer, $\tilde{\psi}$ is the adjacency matrix of the graph with added self-connections, $\tilde{D}$ is the degree matrix of $\tilde{\psi}$, $W^{(l)}$ is the weight matrix for the $l$-th layer, and $\sigma$ denotes a ReLU function.

After refining the graphs and decoupling into inter- and intra-view features by the CHA module, the node attributes are analyzed at the genetic, spatial, and collaborative levels. Subsequently, the resulting triple heterogeneity-aware graphs are globally aligned and mapped to a unified representation as follows:

$$\psi_{spa}^r, \psi_{gene}^r, \psi_{joint}^r = \text{CHA}\left(\psi_{spa}, \psi_{gene}\right),$$
$$\psi_{refine} = \text{UAP}\left(\psi_{spa}^r, \psi_{gene}^r, \psi_{joint}^r\right), \quad (4)$$

where UAP stands for universal aligned projection. The resulting $\psi_{refine}$ is a comprehensive and well-constructed aggregated representation that maximizes multi-view information expression in a confined space. Thus, stGCL not only reveals spatial and genetic associations at the molecular level, but also exhibits sound generalizability across multiple downstream tasks.

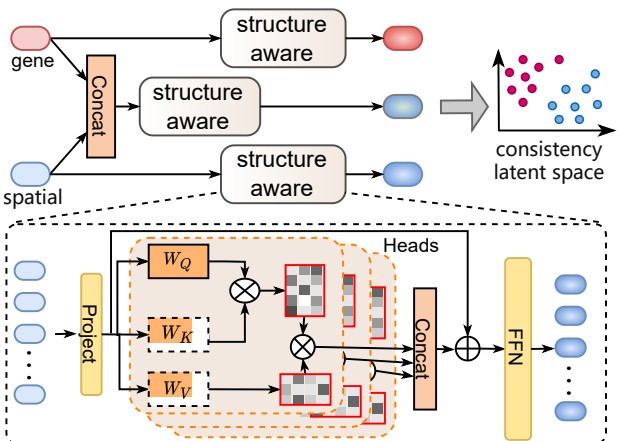

**Figure 3: Structures of cross-view hierarchical feature align module. Decoupling spatial and genetic features by inter- and intra-view feature design, where spatial structure perception uses $W_Q$, $W_K$, and $W_V$ for feature spatial transformations to obtain molecular structural affinities.**

### 3.2 Cross-view Hierarchical Feature Align.

We first show that view heterogeneity informs the aggregation strategy and then analyze it through information theory. After obtaining genetic and spatial graphs with specific features derived from independent parallel encoders, embedding them in joint latent distributions, and identifying cell categories through bi-determined expanded central cluster distributions, can be formulated as: $\psi_{joint} = C\left(\psi_{gene}, \psi_{spa}\right)$. As shown in Fig. 3, we generate three layers of features to extract low-frequency information through dynamic structure-aware attention respectively, which helps to update $\psi_{gene}$ and $\psi_{spa}$ from the prior distribution in the case of $\psi_{joint}$. This process can be expressed as:

$$\psi_{gene}^{n+1} = \psi_{gene}^n + \mathcal{T}\left(\mathcal{A}\left(\psi_{gene}^n\right)\right), \psi_{spa}^{n+1} = \psi_{spa}^n + \mathcal{T}\left(\mathcal{A}\left(\psi_{spa}^n\right)\right)$$

$$\psi_{joint}^{n+1} = \psi_{joint}^n + \mathcal{T}\left(\mathcal{A}\left(\psi_{joint}^n\right)\right),$$

(5)

where $\mathcal{T}$ denotes the transformer block, $C(\cdot)$ indicates channel concat, and the bi-directional attention is denoted as:

$$\mathcal{A}(X) = \text{softmax}\left(\frac{XW_Q \cdot XW_K}{\sqrt{d}}\right) \cdot XW_V,$$

(6)

where $\sqrt{d}$ is the scaling factor and $W$ is the weight transformation matrix. This cross-modality hierarchical decoupling guidance allows the fused image to diffuse out of the constraints of the vanilla distribution, ensuring that it encompasses the low-frequency diverse distributions of both modalities.

**Universal Aigned Project** We then obtain two intra-view spatial and gene features and an inter-view collaborate feature. These are aligned onto a distribution $\psi_{universal}$, which possesses a larger size compared to $\psi_{gene} + \psi_{spa} + \psi_{joint}$, formulated as:

$$\psi_{universal} = \mathcal{P}\left(C\left(\psi_{gene}, \psi_{spa}, \psi_{joint}\right)\right),$$

(7)

where $\mathcal{P}$ comprises a stacked layer of transformer and MLP. Then, $\psi_{refine}$ is condensed into a succinct and complete representation of $\psi_{universal}$, denoted as:

$$\psi_{refine} = \text{UAP}\left(\psi_{universal}\right),$$

(8)

where UAP consists of three layers of MLP. Employing the mapping-to-deflation strategy, we achieve a refined representation $F$ that not only optimally restores spatial and genetic features but also enhances downstream task analyses.

Next, we analyze the traditional fusion paradigm of spatial transcriptomics from an information theory perspective.

**Theorem 1.** In the fusion of multiple views into a compact fusion feature, the mapping to global spatial deflation strategy is better expressed and more effective than direct fusion.

**Proof.** According to information theory [25], consider two non-orthogonal modalities $X$ and $Y$ with sizes $x$ and $y$, respectively, and an aggregate mode $F$ of size $f$. When $f \geq x + y$, a model can be learned that contains all information of $X$ and $Y$, that is,

$$I(F; X, Y) = H(X, Y) - H(X, Y|F),$$

$$= -\sum_{i=1}^{n}\sum_{j=1}^{m} p(x_i, y_j) \log p(x_i, y_j),$$

(9)

where $I$ denotes mutual information and $H$ denotes entropy.

Given the sparsity of gene data, the aggregation process should satisfy $f < x + y$, aiming for a compact model that retains the connection and original characteristics of $X$ and $Y$ within a narrower distribution. Typically, in spatial transcriptomics data, we often find $f = x = y$. The mutual information acquired, $I(F; X) + I(F; Y)$, represents the combined information about $X$ and $Y$ retrievable from $F$. When $f < x + y$, the maximization tends to prioritize the shared information of $X$ and $Y$, denoted as:

$$\max\{I(F; X) + I(F; Y)\},$$

$$\text{where } I(F; X) = \int_F \int_X p(f, x) \log \frac{p(f, x)}{p(f)p(x)} \, df \, dx.$$

(10)

This objective function ignores the orthogonal (independent, non-overlapping) information components, resulting in a biased representation of $F$ towards either $X$ or $Y$, leading to an inconsistent expression of the combined modality.

Considering the heterogeneity of spatial and gene data in ST, the joint entropy $H(X, Y)$ is minimal and difficult to aggregate, complicating the creation of a compact, efficient, and effective representation. This complexity becomes especially apparent in various downstream tasks, where a task might rely on the independent information components of $X$ or $Y$, potentially causing the learned modal aggregation information to collapse. As a result, the learned $Z$ may lack the ability to adequately express $X$ and $Y$.

Inspired by archery in Fig. 1, our method leverages the prior distributions $P(T|(X, Y))$ of $X$ and $Y$ through an intermediate distribution $T$ of size $t$, where $t > x + y > f$, formulated as:

$$H(T) = H(X) + H(Y) + H(A),$$

(11)

where $X, Y, A, T$ corresponds to $\psi_{gene}, \psi_{spa}, \psi_{joint}$ and $\psi_{universal}$ in the CHA module of Section 3.2, indicating that $T$ captures information from both $X$ and $Y$. Subsequently, the information within $T$ undergoes low-pass filtering, and $T$ is compressed through a low-pass filter to obtain unbiased estimates $F$ of $X$ and $Y$. This

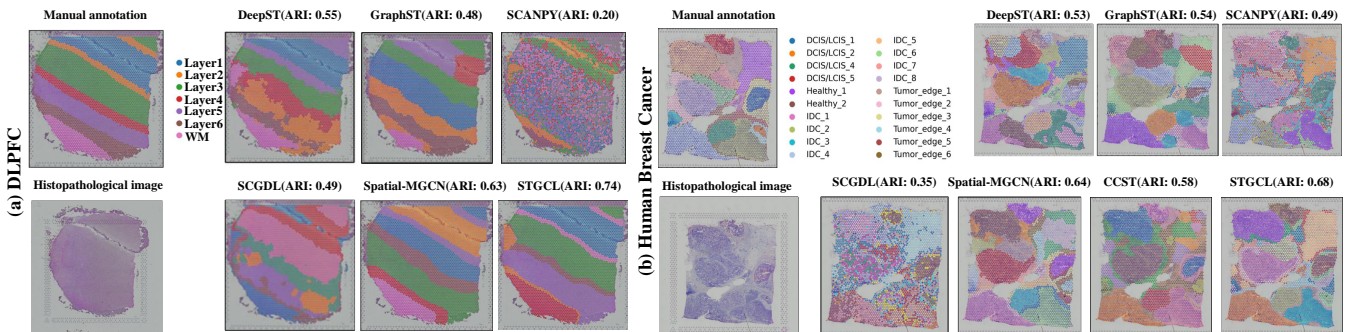

**Figure 4: Identify spatial domains for comparison experiments between DLPFC and Human Breast Cancer datasets. The manual annotation of the slice # 151507 in DLPFC.**

strategy enables $F$ to learn multi-view orthogonality and shared essential features by aggregating $X$ and $Y$ through $T$ rather than directly through $F$, thereby facilitating the discovery of the alignment mapping of $X$ and $Y$ within $F$. More details are provided in the Supplementary Material.

## 3.3 Contrastive Heterogeneous Molecular Learning.

The spatial maps of cell populations exhibit high edge similarity, but their type discrimination relies solely on genetic features, resulting in spatial features blurring tissue boundaries. Therefore, we categorize the obtained views into triple levels, synergistically identifying cells in parallel. At the collaborative level, we distinguish nodes identified as the same type by multiple views by comparing the loss of cross-view features, thereby minimizing the disparity between gene and spatial features. The loss can be defined as:

$$\mathcal{L}_{cross} = -\sum_{i=1}^{N} \sum_{j=1}^{M} w_{ij} \left[ y_{ij} \log \left( \sigma(\psi_{spa}^i, \psi_{gene}^j) \right) \right. \\ \left. + \left( 1 - y_{ij} \right) \log \left( 1 - \sigma(\psi_{spa}^i, \psi_{gene}^j) \right) \right], \quad (12)$$

where $N$ and $M$ represent the spatial and gene data sample sizes, respectively. $s_i$ and $g_j$ represent the feature representations of the $i^{th}$ spatial data sample and the $j^{th}$ gene data sample. $y_{ij}$ serves as a binary label indicating whether samples $i$ and $j$ belong to the same category (e.g., $y_{ij} = 1$ for same category, and $y_{ij} = 0$ for different categories). $w_{ij}$ dynamically assigns weights to spatial and genetic information, using intra-view features (i.e., genes) to alleviate category mixing issues at cell boundary locations. This process leverages view heterogeneity relationships to drive $\psi_{textgene}$ and $\psi_{textspa}$ closer together, guiding the cross-view feature $\psi_{textjoint}$ to effectively differentiate between nodes universally recognized as being of the same kind across multiple views.

As a single spot may represent the mean gene expression of multiple cells at the gene level, we employ the zero-inflated negative binomial distribution [32] to capture crucial aspects of gene expression data, including zero inflation (high sparsity) caused by true and dropout zeros, discreteness, and over-dispersion (variance greater than the mean). We model the distribution $p(\psi_{gene} \mid Z)$ as:

$$p(\psi_{gene} \mid Z) = \prod_{i=1}^{n_s} p\left( \hat{\psi}_i \mid z_i \right) = \prod_{i=1}^{n_s} p\left( \hat{\psi}_i \mid \pi_i, \mu_i, \sigma_i^2 \right), \quad (13)$$

where $p\left( \hat{\psi_{gene_i}} \mid \pi_i, \mu_i, \sigma_i^2 \right)$ is the ZINB distribution parameterized by $\pi_i, \mu_i, \sigma_i^2 \in \mathbb{R}^{n_g}$. Specifically, $\pi_i = \text{sigmoid} \left( W\pi f_D^{(2)}(z_i) \right)$ represents the zero rate vector of this distribution, where $W_\pi$ signifies the weight matrix of $\pi$. $\mu_i = \exp \left( W_\mu f_D^{(2)}(z_i) \right)$ denotes the mean of the associated negative binomial, with $W_\mu$ as the weight matrix of $\mu$. $\sigma^2 = \exp \left( W_\sigma f_D^{(2)}(z_i) \right)$ represents the variance of the negative binomial, where $W_\sigma$ is the weight matrix of $\sigma^2$. The loss function of the parameter estimation is defined as the negative log-likelihood of the ZINB distribution:

$$\mathcal{L}_{zinb} = -\log(p(\psi_{gene} \mid \hat{\pi}, \hat{\theta}, \hat{\mu})). \quad (14)$$

On the other hand, spatially adjacent spots should be proximate, whereas spatially non-adjacent spots should be distant in the latent space. Thus, we incorporate both similarity and spatial neighbor features to compute the spatial regularization constraint loss:

$$\mathcal{L}_{reg} = -\sum_{i=1}^{n} ( \sum_{j \in \mathcal{R}_i} \log \left( \sigma \left( \psi_{ij} \right) \right) + \sum_{k \notin \mathcal{R}_i} \log \left( 1 - \sigma \left( \psi_{ik} \right) \right)), \quad (15)$$

where $n$ represents the number of spots, this method minimizes the embedding distance between spatial neighbor spots through the spatial regularization constraint, enhancing the informativeness and discriminative power of the learned latent space.

**Overall loss function.** During the training process, the multi-view GCN encoder, the ZINB decoder, and the spatial regularization constraints are collectively optimized. The ultimate training objective of stGCL is defined as:

$$\mathcal{L}_{total} = \alpha \mathcal{L}_{con} + \beta \mathcal{L}_{zinb} + \gamma \mathcal{L}_{reg} + \lambda \mathcal{L}_{cross}, \quad (16)$$

where $\alpha$, $\beta$, $\gamma$, and $\lambda$ represent the weighting factors used to balance the influences of reconstruction loss, consistency loss, and spatial regularization constraint loss.

## 4 EXPERIMENTAL RESULTS

### 4.1 Datasets and Details

The **DLPFC** [15] dataset comprises 12 tissue slices obtained from 3 adult samples sourced from the dorsolateral prefrontal cortex of individuals from the Lieber Institute for Brain Development (LIBD), each containing four adjacent slices. These slices, obtained using 10x Visium, were manually labeled to identify DLPFC layers and

**Table 1: Comparison of ARI metrics between stGCL and other spatial transcriptomics approaches on DLPFC, Human Breast Cancer, and Mouse Brain Anterior Tissue datasets. The 12 datasets in DLPFC are indicated by number. Boldface and under-line show the best and second-best values, respectively.**

| Datasets | DeepST | GraphST | SCANPY | SCGDL | SpaGCN | Spatial-MGCN | stGCL |
|---|---|---|---|---|---|---|---|
| 151507 | 0.55 | 0.48 | 0.20 | 0.49 | 0.43 | 0.63 | **0.74** |
| 151508 | 0.42 | 0.49 | 0.15 | 0.34 | 0.33 | 0.46 | **0.53** |
| 151509 | 0.43 | 0.52 | 0.19 | 0.32 | 0.41 | 0.54 | **0.67** |
| 151510 | 0.50 | 0.50 | 0.14 | 0.31 | 0.37 | 0.51 | **0.73** |
| 151669 | 0.44 | **0.48** | 0.10 | 0.24 | 0.23 | 0.39 | 0.44 |
| 151670 | 0.33 | 0.46 | 0.09 | 0.26 | 0.21 | 0.35 | **0.48** |
| 151671 | 0.52 | 0.61 | 0.12 | 0.31 | 0.34 | 0.60 | **0.67** |
| 151672 | 0.48 | 0.63 | 0.12 | 0.34 | 0.38 | 0.77 | **0.82** |
| 151673 | 0.54 | **0.63** | 0.20 | 0.33 | 0.40 | 0.61 | 0.57 |
| 151674 | 0.58 | 0.43 | 0.22 | 0.29 | 0.31 | 0.60 | **0.62** |
| 151675 | 0.43 | 0.55 | 0.23 | 0.24 | 0.33 | 0.54 | **0.56** |
| 151676 | 0.54 | **0.61** | 0.22 | 0.21 | 0.28 | 0.58 | 0.58 |
| Human Breast Cancer | 0.53 | 0.54 | 0.49 | 0.35 | 0.56 | 0.64 | **0.68** |
| Mouse Brain Anterior | 0.25 | 0.41 | 0.23 | 0.26 | 0.32 | 0.42 | **0.45** |

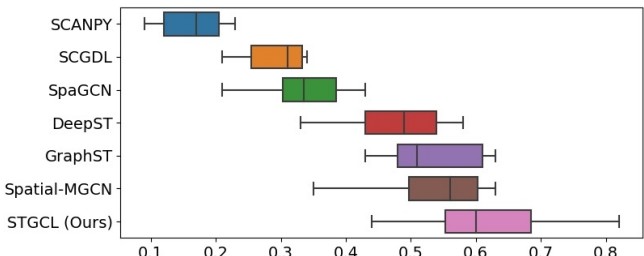

**Figure 5: Boxplots of ARI values for seven methods across 12 slices of DLPFC.**

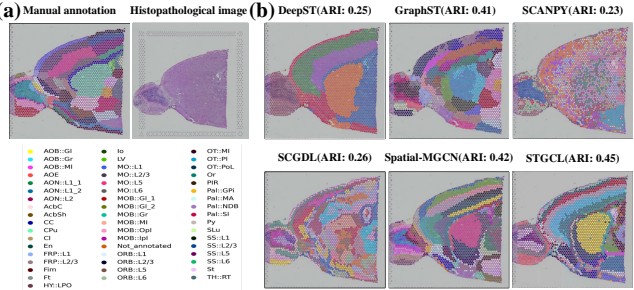

**Figure 6: Identify spatial domains on Mouse Brain Anterior Tissue dataset. (a) Manual annotation layer structure and the histopathological image for human breast cancer dataset. (b) Spatial domains are detected with stGCL and five methods.**

white matter (WM). Within each slice, there were typically five to seven manually labeled regions. The **Human Breast Cancer** dataset includes the 10X Visium dataset for human breast cancer [1]. This dataset has been meticulously annotated into 20 distinct regions, categorized into four principal morphological types: ductal

carcinoma in situ/lobular carcinoma in situ (DCIS/LCIS), invasive ductal carcinoma (IDC), healthy tissue areas, and hypo malignant tumor margins. The **Mouse Brain Anterior Tissue** dataset was annotated with 52 different regions.

**Implementation Details and Metrics.** The complete model is implemented using PyTorch version 1.12.1. For the training phase, we set the learning rate to 0.001 and applied a weight decay 0.0005. We conducted 200 training rounds in each experiment conducted on the DLPFC, Human Breast Cancer, and Mouse Brain Anterior Tissue datasets. For validation of spatial clustering, we employ the Adjusted Rand Index (ARI) as our evaluation metric. All experiments were executed on an NVIDIA RTX 3090 GPU.

## 4.2 Comparison with SOTA methods

To demonstrate the superior performance of the proposed stGCL, we have chosen six benchmark methods for comparative analysis, including DeepST [29], GraphST [14], SCANPY [28], SCGDL [12], SpaGCN [9], and Spatial-MGCN [26].

*4.2.1 DLPFC dataset.* Comparative tests between stGCL and six other models across 12 DLPFC slices are presented in Table 1. Our method outperforms the existing state-of-the-art on nine of these datasets, achieving performance enhancements of over 5% in clusters ranging from 151507 to 151672, particularly with a 22% improvement on #151510. Notably, many methods exhibit extreme performance dips below 30% on specific datasets, which significantly undermines the reliability and applicability of results in both scientific research and clinical settings. In contrast, stGCL surpasses the average performance of competing methods across all datasets, with several clusters showing improvements above 40%, thereby demonstrating its robustness and broad applicability.

Fig. 4a illustrates the clustering visualization results for sample #151507. The stGCL algorithm effectively separates cell clusters and clearly outlines tissue boundaries in elongated structures, where spatial information is closely related. This result highlights the value

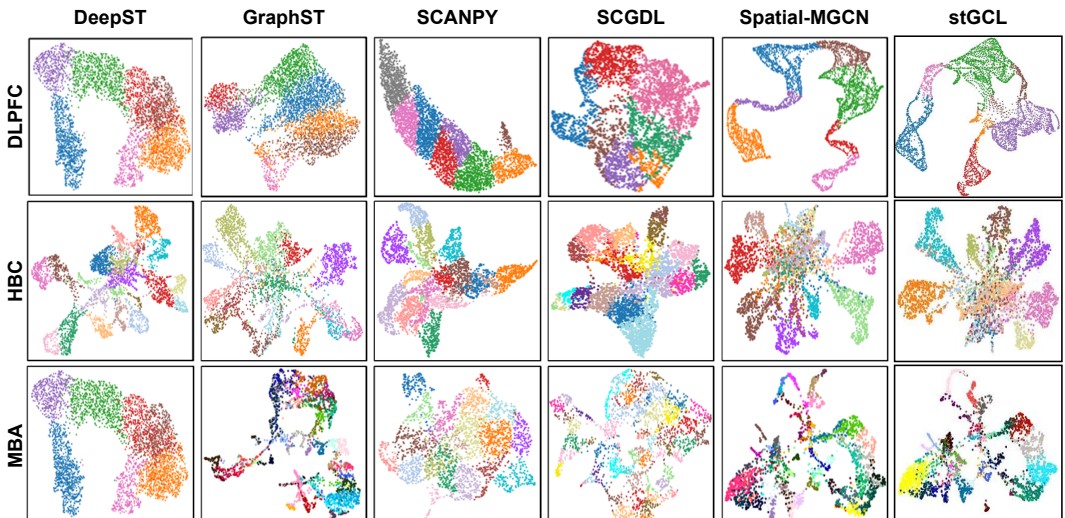

**Figure 7: The UMAP visualization of these results. UMAP indicates uniform manifold approximation and projection.**

of stGCL's hierarchical processing of ST data. Moreover, Fig. 5 showcases the overall performance of the algorithms, clearly indicating that stGCL performs better than the competing algorithms.

*4.2.2   10x Visium Human Breast Cancer dataset.* Table 1 compares our method with six other models. Although clustering performance among these models is generally similar due to significant differences in the sizes of small cell populations across 21 categories, stGCL achieves a 4% improvement over the highest-performing model, Spatial-MGCN, with an ARI of 68%. This indicates that stGCL effectively addresses the challenges of category imbalance between large and small cell clusters by leveraging heterogeneous comparisons to identify ambiguous nodes.

Cluster visualization displays the situation of the human breast cancer dataset, as shown in Figure 4b. Unlike SCGDL, DeepST, and SCANPY, stGCL not only misclassifies fewer nodes but also delineates small cell populations. Although most methods produce visualizations with irregularly shaped cells and usually display new negative sample categories in the center of cell clusters, particularly in the top right and top left corners, stGCL successfully clarifies the tissue shapes. This method enhances the distinction at cell boundaries by shifting focus from inter-view to intra-view features, leading to more accurate node classification.

*4.2.3   Mouse Brain Anterior Tissue dataset.* Fig. 6 6 and the last row of Table 1 display the comparative results between stGCL and other algorithms. Notably, stGCL uniquely and accurately delineates the layer structure, achieving the highest ARI of 0.45, surpassing the performance of alternative methods. In contrast, competing algorithms such as DeepST, SCANPY, and SCGDL report lower ARI scores of 0.25, 0.23, and 0.26, respectively. These methods tend to amalgamate recognized layers, failing to represent the trustworthy structural organization of the data accurately. This evidence indicates that stGCL offers a more precise and dependable analysis of ST data, enhancing the utility of spatial and genetic features through optimized mutual information between views.

**Table 2: Ablation study measured on DLPFC dataset. The ARI metric uses the average of all datasets. SA denotes structure-aware module, and UAP denotes universal aligned projection.**

| Methods | CHA | SA | UAP | $\mathcal{L}_{cross}$ | ARI |
|---|---|---|---|---|---|
| Baseline | - | - | - | - | 0.5175 |
| w/o $\mathcal{L}_{cross}$ | ✔ | ✔ | ✔ | - | 0.5864 |
| w/o CHA | - | - | - | ✔ | 0.5519 |
| w/o UAP | - | ✔ | - | ✔ | 0.5643 |
| w/o SA | - | - | ✔ | ✔ | 0.5946 |
| stGCL | ✔ | ✔ | ✔ | ✔ | 0.6175 |

A comprehensive analysis in Table 1 demonstrates that stGCL exhibits state-of-the-art performance across various datasets, including DLPFC, human breast cancer, and mouse brain anterior, showcasing effective generalization.

## 4.3   Ablation studies

In this study, we conducted an ablation experiment using DLPFC to evaluate the contributions of various components within our proposed model. The results, presented in Table 2, elucidate the individual and collective impacts of the Structure-Aware module (SA), universal aligned projection (UAP) module, and cross-entropy loss function ($\mathcal{L}_{cross}$) on the performance, as measured by the ARI.

**Effect of CHA module.** To verify the validity of our framework and information theory, we remove the SA and UAP components of CHA, respectively. It can be seen that w/o SA is 2% lower than stGCL, indicating that global structure awareness is helpful for highly sparse gene data.

**Effect of UAP.** Analysis of the third and fifth rows in Table 1 indicates that incorporating UAP enhances performance by over 4%, further validating the effectiveness of our mapping to deflation strategy. Additionally, removing the UAP component while maintaining SA and $\mathcal{L}cross$, the ARI of 0.5643 is 3% lower than w/o SA.

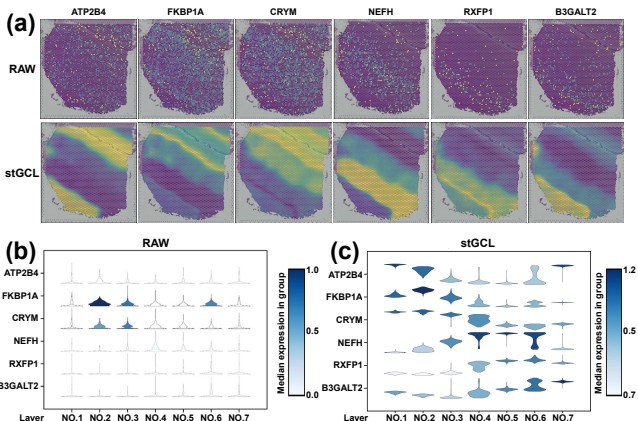

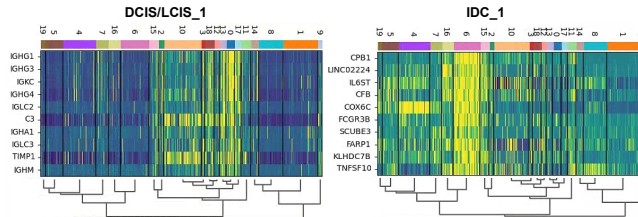

Figure 8: (a) The visualization of raw expression of layer marker genes and expression after stGCL imputation. (b) The violin plots of raw cortical marker gene expression and stGCL cortical marker gene expression. (c) The violin plot of cortical marker gene expression imputed by stGCL.

This suggests that while the structure-aware module contributes to model validity, its impact is not as critical as that of the UAP, as evidenced by the higher ARI of 0.5946 obtained for the model without SA, including UAP and $\mathcal{L}_{cross}$. Hence, the UAP module from mapping to deflation is more effective than the SA module's low-pass filter-like fusion paradigm, maximizing the retention of essential information in multiple view features.

**Effect of $\mathcal{L}_{cross}$.** As shown in the first and third rows of Table I, independently incorporating $\mathcal{L}_{cross}$ leads to a 4% improvement, proving that heterogeneity-based guided comparison learning is indispensable. On the other hand, w/o $\mathcal{L}_{cross}$ and w/o CHA with stCGL showed a performance difference of 3% and 6%, respectively. This indicates that the CHA module and $\mathcal{L}_{cross}$ are complementary, synergistically enhancing multi-view heterogeneity learning.

### 4.4 Downstream applications

*4.4.1 UMAP visualization.* We illustrate the uniform manifold approximation and projection (UMAP) visualization results of the stGCL method alongside five other methods applied to three datasets, as depicted in Fig. 7. stGCL accurately delineates the orderly development of individual cortical layers, including layers 1 through 6 and white matter (WM), outperforming the baseline methods. Interestingly, the UMAP plots of most methods (e.g., GraphST, SCGDL) exhibit insufficient point spacing between layers, which blurs the distinction between them. In contrast, stGCL achieves greater separation between clusters on all three datasets and exhibits a more structured expansion in various directions, demonstrating how heterogeneity modeling effectively enhances the differentiation between categories by synergizing spatial and genetic clustering. Compared to Spatial-MGCN, the class clusters of stGCL are more compact, with branches more precisely arranged within the constrained space. This improvement stems from our approach of employing comparative learning to effectively mitigate confounding factors in the slightly mixed developmental trajectories typical

Figure 9: The Heatmap of the expression of the structural domains on the top 10 DEGs between Healthy 1 and DCIS/LCIS.

of methods such as SCANPY. At the same time, stGCL's embeddings display the expected cortical layer structure. Consequently, stGCL enhances the spatial domain recognition of ST data while preserving essential biological features.

*4.4.2 Gene Imputation.* Spatial transcriptomic data are frequently compromised by noise and data loss, adversely affecting the accuracy of gene expression analyses. To establish that stGCL employs heterogeneity-guided learning to preserve critical tissue distribution information from raw data efficiently, we introduced this method, which utilizes positive and negative sample comparisons guided by tissue heterogeneity. When applied to DLPFC, stGCL enabled analysis of the spatial expression patterns of six primitive layer marker genes (ATP2B4, FKBP1A, CRYM, NEFH, RXFP1, B3GALT2). Fig. 8 illustrates how stGCL mitigates considerable noise interference in slice #151507 of DLPFC raw data. The stGCL interpolation generates embeddings that accurately delineate cortical layer boundaries.

These findings demonstrate that stGCL effectively eliminates irrelevant noise and data loss artifacts, and dynamically captures and reconstructs spatial transcriptomics data. This ensures that the spatial distributions of these marker genes are consistent with prior observations. Comparative analyses using violin plots of raw and interpolated gene expressions reveal significant improvements in spatial expression patterns, which more closely align with manually annotated organizational structures. Hence, stGCL excels in interpolating gene expression, demonstrating its superior ability to maintain critical spatial data integrity.

Furthermore, to further explore the heterogeneity of tumor tissue, we focused on the DCIS/LCIS_1 and IDC_1 clusters to analyze the expression of the top 10 DEGs shown in Fig. 9. This analysis revealed significant heterogeneity among the clusters.

## 5 CONCLUSIONS

We introduce the heterogeneous graph guided contrastive learning (stGCL) method for aggregated ST data. This method effectively merges gene and spatial information into unified potential distributions by dynamically refining node attributes across various levels, leveraging cross-view heterogeneity. The cross-view hierarchical feature align module ensures structural integrity and optimal feature aggregation, using information principles. Experimental evaluations reveal that stGCL outperforms current methods in various tasks, showcasing its potential in complex biological analyses through advanced contrastive learning.

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
