# OpenReview forum: "Heterogeneous Graph Guided Contrastive Learning for Spatially Resolved Transcriptomics Data"
_acmmm.org/ACMMM/2024/Conference — MM2024 Poster_

### Official Review · Reviewer_Wpat · 2024-04-29

**Rating:** 5
**Confidence:** 3

**Summary:**

The paper introduces an interesting approach called Heterogeneous Graph Guided Contrastive Learning (stGCL) for analyzing spatially resolved transcriptomics data. This method leverages gene sequencing and spatial imaging technologies to explore cellular interactions and disease mechanisms. The stGCL framework incorporates three levels of node attributes based on spatial and genetic heterogeneity relationships, enabling unbiased refinement through contrastive learning loss. It also includes a cross-view hierarchical feature alignment module to separate spatial and genetic views while aggregating features, enhancing the integrity of inter- and intra-views. Experimental results demonstrate the superiority of stGCL over existing methods in various tasks and downstream applications in the field of spatial transcriptomics.

**Strengths:**

+ The paper effectively communicates the related concepts and framework in a clear and understandable manner, making it accessible to a wide audience.
+ The flow of the paper is well-structured, with logical progression from introduction to methodology and results, enhancing readability and comprehension.
+ The studied topic, aiming to learn multi-level representations for spatial transcriptomics analysis, is indeed critical and could bring far-reaching real-world impacts.
+ The idea to incorporate the heterogeneity of genetic and spatial a priori distributions is technically sound and inspring.

**Limitations:**

- Only one evaluation metric is adopted during experimental comparisons. As previously raised by the authors, the learned representations could be used for various down-stream tasks. However, no quantitative results are presented in those aspects. It would be appealing to see if the proposed graph contrastive learning method could attain good results on those tasks.
- The proof for theorem 1 is not rigorous. Currently, the deduction is mostly qualitative and intuitive, not mathematically sound.

**Suitability:**

3

---

### Official Review · Reviewer_iRUe · 2024-05-19

**Rating:** 4
**Confidence:** 3

**Summary:**

The authors presents a method called stGCL for analyzing spatial transcriptomics data. The approach leverages the inherent heterogeneity of spatial and genetic information by dynamically coordinating triple-level node attributes through comparative learning loss. The introduction of a cross-view hierarchical feature alignment module further enhances the integrity of both spatial and genetic views. Experiments demonstrate that stGCL outperforms existing methods in various tasks and related downstream applications.

**Strengths:**

1.  The proposed stGCL method addresses the challenge of integrating spatial and genetic information by maintaining view independence during aggregation. The CHA module maximizes the preservation of original molecular information, enhancing the interpretability and accuracy of the results.

2.  The authors conducted extensive experiments across multiple datasets, demonstrating the quantitative and qualitative superiority of their method. This thorough evaluation provides strong evidence for the effectiveness of stGCL.

**Limitations:**

1.  The method is pretty complex. And some sections detailing the methodology are densely packed with information, such as the collaborate-level.

2.  The scalability of the stGCL method to very large datasets is not thoroughly addressed. Given the increasing size of spatial transcriptomics datasets, a discussion on computational efficiency and scalability would be valuable.

Minor:
3. How were the hyperparameters of the weighting of the 4 training objectives tuned to make the method work well? The inclusion of ablation studies to analyze the impact of different weighting of loss function is also needed.

**Suitability:**

2

---

### Official Review · Reviewer_wDN5 · 2024-05-26

**Rating:** 4
**Confidence:** 3

**Summary:**

In this study, the authors propose a method to learn representations from different views for spatial transcriptomics data. The key elements of their approach include heterogeneous graph, contrastive learning and  feature alignments.

**Strengths:**

1. This paper is easy to read and the expression is clear.
2. The method presented in the paper appears to be technically sound.
3. The experiments are extensive and comprehensive.

**Limitations:**

1. Is it necessary to construct graph structured data or heterogeneous graph structured data?  How can we ensure the accuracy of the constructed graph data, and that the connections can reflect the relationships in the biological information?
2. The authors directly use the expression X of genes to calculate similarity, which I think is unreasonable. Firstly, the expression is high-dimensional, and the distance does not represent the biological relationship. Secondly, if the original expressions of these samples are already similar (A_ij), what is the meaning of using GCN? In my opinion, there are many defects and issues in the method.)
3. I feel that the clinical significance of this paper is not very high. Specifically, in the fields of bioinformatics analysis and medical data analysis, the most important aspect is the clinical significance, followed by the methods and data.
4. The method lacks innovation, and the theoretical content is also not very useful. Most of the information theory-based method can be written in a similar way.

**Suitability:**

2

---

### Official Review · Reviewer_yJ8T · 2024-06-10

**Rating:** 4
**Confidence:** 2

**Summary:**

This paper studies a problem of spatial transcriptomics. Specifically, this paper proposes a method named stGCL (spatial transcriptomics Graph Contrastive Learning) to analyze spatial transcriptomics data. This method aims to enhance the understanding of cellular interactions and disease mechanisms by leveraging a contrastive learning framework that respects the heterogeneity of spatial and genetic data. The experiments on DLPFC, Human Breast Cancer and Mouse Brain Anterior Tissue  datasets demonstrate the effectiveness of stGCL over baselines.

**Strengths:**

1. The topic that studies spatial transcriptomics is practical and interesting.
2. The experimental results  of stGCL on 3 real-world datasets are good.
3. The idea to integrating spatial and genetic data is interesting.

**Limitations:**

1. Since this paper consider two modalities, gene and spatial. I would like to know the contribution of each modality. Would you have discuss theoretically and empirically?
2. Why not just directly apply some existing method for heterogeneous graphs, e.g., [1]? Could you have a discussion about it?


[1] Heterogeneous Graph Neural Network. KDD 2019.

**Suitability:**

2

---

### Meta-Review · Area_Chair_EeGj · 2024-06-27

**Recommendation:** Accept (Poster)
**Confidence:** 4

**Metareview:**

According to all the review comments, rebuttals, discussions and final ratings, the majority of the reviewers gave positive ratings to this paper and the concerns were well addressed. I am happy to recommend to accept this paper. Please carefully revise the final manuscript according to the comments and discussions.